# *Ex vivo* model of herpes simplex virus type I dendritic and geographic keratitis using a corneal active storage machine

Emilie Courrier[1], Corantin Maurin[1], Victor Lambert[2], Didier Renault[1], Thomas Bourlet[3], Sylvie Pillet[3], Paul O. Verhoeven[3], Fabien Forest[4], Chantal Perrache[1], Zhiguo He[1], Thibaud Garcin[1,2], Antoine Rousseau[5,6], Marc Labetoulle[5,6], Philippe Gain[1,2], Gilles Thuret[1,2,7] *

1 Corneal Graft Biology, Engineering and Imaging Laboratory, Health Innovation Campus, Faculty of Medicine, Jean Monnet University, Saint-Etienne, France, 2 Department of Ophthalmology, University Hospital, Saint-Etienne, France, 3 Laboratory of Infectious Agents and Hygiene GIMAP-EA3064, University Hospital & University Jean Monnet, Saint-Etienne, France, 4 Department of Pathology, University Hospital, Saint-Etienne, France, 5 Department of Ophthalmology, Bicêtre Hospital, APHP, South Paris University, Le Kremlin-Bicêtre, France, 6 Center for Immunology of Viral Infections and Autoimmune Diseases, IMVA, UMR, INSERM, CEA, South Paris University, Fontenay-aux-Roses, France, 7 Institut Universitaire de France, Paris, France

* gilles.thuret@univ-st-etienne.fr

**Data Availability Statement:** All relevant data are within the manuscript and its Supporting Information files.

## Abstract

### Background

Herpetic keratitis (HK) models using whole human corneas are essential for studying virus-host relationships, because of high species specificity and the role of interactions between corneal cell populations that cell culture cannot reproduce. Nevertheless, the two current corneal storage methods (hypothermia and organ culture (OC)) do not preserve corneas in good physiological condition, as they are characterized by epithelial abrasion, stromal oedema, and excessive endothelial mortality.

### Methods

To rehabilitate human corneas intended for scientific use, we used an active storage machine (ASM) that restores two physiological parameters that are essential for corneal homeostasis: intraocular pressure and storage medium renewal (21mmHg and 2.6 µL/min, respectively). ASM storage regenerates a normal multilayer epithelium in 2 weeks. We infected six pairs of corneas unsuitable for graft by inoculating the epithelium with herpes simplex virus type 1 (HSV-1), and compared each ASM-stored cornea with the other cornea stored in the same medium using the conventional OC method.

### Results

Only corneas in the ASM developed a dendritic (n = 3) or geographic (n = 2) epithelial ulcer reproducing typical HSV-1-induced clinical lesions. Corneas in OC showed only extensive desquamations. None of the uninfected controls showed epithelial damage. Histology,

**Funding:** The authors received no specific funding for this work.

**Competing interests:** Please note that PG and GT are inventors of the active storage machine on "patent US 20160029618A1" submitted by Jean Monnet University that covers "Medical device intended for long-term storage of a cornea, or for ex vivo experimentation on a human or animal cornea". The other authors have no proprietary or commercial interest in any materials discussed in this article."I can confirm that this does not alter our adherence to PLOS ONE policies on sharing data and materials.

immunohistochemistry, transmission electron microscopy and polymerase chain reaction on corneal tissue confirmed infection in all cases (excluding negative controls).

## Conclusions

The ASM provides an innovative *ex vivo* model of HK in whole human cornea that reproduces typical epithelial lesions.

## Introduction

Herpes simplex virus type 1 (HSV-1) has a particular tropism for human sensory neurons of the upper body. HSV keratitis (HK) is the leading cause of infectious blindness in Western countries [1]. The prevalence of ocular HSV infection was estimated in 1980s America at 149 cases per 100,000 individuals with an incidence of 8.4 new cases per 100,000 individuals per year [2]. These data are similar to the annual incidence of all types of new ocular HSV infection, estimated in a more recent study at 11.8 per 100,000 people in a similar population [3]. In France, the incidence was in 2005 estimated at 13.2 per 100,000 person-years for new cases and 18.3 per 100,000 person-years for recurrences [4].

Excluding neonatal infections, more than 95% of HSV eye infections are caused by HSV-1 [1, 5]. After an oral primary infection in early childhood, HSV-1 establishes a life-long latency in the trigeminal ganglia, where no viral progeny but abundant viral RNAs, referred to as latency-associated transcripts, are produced [6]. Certain circumstances (fever, ultraviolet exposure, immunosuppression, nerve injuries) can cause viral reactivation (recurrence) in which new viral particles are produced and emerge unilaterally in the tissues innervated by the trigeminal ganglion, often including the ocular surface and periocular tissues [6, 7]. Recurrence varies in frequency between subjects, and can cause irreversible damage ranging from opacities to major complications such as neurotrophic keratitis [8].

The diagnosis of HK is most often clinical, based on the presence of typical unilateral corneal lesions: epithelial damage in 66% of cases (dendritic keratitis in 56%, geographic ulcer in 10%) and stromal damage in 30%. The remaining 4% concerns other lesions such as punctate keratitis and ulcer [4]. Typical epithelial lesions form rapidly from small foci of punctate keratitis that coalesce to form a linear ulcer with several branches with rounded ends. The epithelium located on the margins of the ulcer is edematous and therefore slightly prominent. It contains the replicative, cytotoxic virus. Some ulcers, by contrast, expand centrifugally, resulting in a geographic pattern [5].

Although HSV has for decades been known to be responsible for ocular infections [9, 10], studies of virus-natural host interactions using relevant *ex vivo* models are necessary to understand which host factors are important for establishment and control of infection but also to screen novel antiviral drugs.

Three types of HK model exist. Animal experimentation aside, *in vitro* models use cultured immortalized epithelial cell lines from monkey kidney [11, 12] or primary human corneal epithelial cells [13]. They are simple, repeatable, fast, and relatively inexpensive. *Ex vivo* models use fresh or stored human [14–18] or animal corneas [16, 19–21] infected by direct contact of the virus on the epithelium. In theory, these are more physiological than epithelial cultures as they retain some epithelial-stromal interactions. They are also less expensive than animal experiments. Nevertheless, to our knowledge, the macroscopic aspect of the epithelial lesions obtained in these *ex vivo* models has only been described once, so these models cannot be said

to reproduce typical dendritic or geographic lesions. This one exception is a recent model of fresh porcine corneas, where dendritic ulcers formed in a virus inoculum-dependent manner, eventually leading to the formation of larger, geographic ulcers [22].

We recently described an active storage machine (ASM) that dramatically improves the long-term storage of human corneal grafts [23, 24]. By restoring the equivalent of intraocular pressure (15–21 mmHg) and by renewing the storage medium, i.e. two parameters necessary for cornea physiology, the ASM improved survival of endothelial cells and their expression of $Na^+/K^+$ATPase, control of stromal thickness, and maturation of epithelial cells. In a porcine version, the ASM stores pig cornea for 7 days without excessive swelling and with good epithelial and endothelial survival, unlike what is always observed with passive storage methods (simple immersion in culture medium) [25]. In the present study, we used this ASM to create a new HK model in human corneas and compared it with a standard organ culture (OC) system.

## Materials and methods

### Study design

The purpose of the study was to investigate whether human corneas discarded from cornea banks could be improved by an innovative storage method that restores intraocular pressure and culture medium flow, and then serve to create an *ex vivo* model of epithelial HK that reproduces the usual clinical characteristics (dendritic or geographic ulcers). We inoculated a suspension of $10^6$ plaque-forming units (PFU)/mL of HSV-1 (strain HF, ATCC VR-260) on the epithelial surface of two groups of six corneas, one preserved in conventional OC, the other in an ASM that we developed for eye banking and preclinical experimentation.

To obtain the best possible comparability between groups, only pairs of corneas were used and the corneas were randomized between groups. Development of epithelial infection was monitored daily by direct observation. Immunolabeling, histology, electron microscopy and PCR were interpreted blind from the storage group.

### Corneal active storage machine

The ASM (machined in polyether ether ketone) was designed to maintain a sterile closed environment enabling long-term storage of the whole cornea as previously described [23, 24]. Briefly, the cornea was tightly secured to the ASM base, using the scleral rim as a watertight seal, compressed by a clamping ring, to separate the epithelial and endothelial chambers; each chamber was continuously connected by the same culture medium (CM). A peristaltic pump controlled by a pressure sensor and a microcontroller continuously renewed the CM at a rate of 2.6 μL/min, while creating a pressure 21 mmHg higher than atmospheric pressure in the endothelial chamber. The CM was a minimum essential medium-based corneal OC medium containing 2% fetal calf serum (CorneaMax; Eurobio, Les Ulis, France). Two optical-quality transparent windows, either side of the cornea, enabled optical control during experiments. The whole system was placed in a 31˚C dry 5% $CO_2$ incubator because, in this version of the ASM containing gas-permeable silicon tubing, the CM was buffered with bicarbonate.

### Human corneal tissue

Thirteen human corneas (six pairs + 1 cornea) unsuitable for transplantation (endothelial cell density <2000 cells/mm², but >800 cells/mm²) from the eye banks of Saint-Etienne and Besançon were used after informed consent of the relatives, as authorized by French bioethics laws. A further eight human corneas (four pairs) were provided by the anatomy department of

the Saint-Etienne Faculty of Medicine through body donation to science. All procedures conformed to the tenets of the Declaration of Helsinki for biomedical research involving human subjects. Mean donor age was $81 \pm 13$ years (48–95) and mean time from death to procurement was $12 \pm 7$ hours (2–23 hours).

## HSV-1 infection of human corneal tissue *ex vivo*

For each pair of corneas, one was randomized to the control group, the other to the experimental arm, where it underwent a first step of epithelial rehabilitation for 2 weeks in the ASM to regenerate a multilayered epithelium [25] (Fig 1A). The other cornea simply remained immersed in a bottle of the same medium (same batch) for the same time. This duration was chosen because it had previously allowed regeneration of 5–7 layers of a fully mature epithelium in most corneas previously stored in OC. A central linear epithelial abrasion was made with a 30-gauge needle and the corneal surface was incubated with 500µL of culture medium containing a suspension of $10^6$ PFU/mL of HSV-1 (strain HF, ATCC VR-260). Stock titers were determined by plaque assay on fibroblastic cell HEL 299 (ATCC CCL-137) monolayers. The infected medium was only placed in contact with corneal epithelium to avoid direct infection of the endothelium: for the ASM group, the medium was simply poured into the epithelial chamber (Fig 1B); for the OC group, the cornea was placed epithelium down in a concave holder containing the medium. Both corneas were then incubated at 37˚C and 5%$CO_2$ for 1 hour, then triple-rinsed with sterile phosphate buffered saline (PBS). The OC cornea was returned to a new vial of CM, and in the ASM group the lid was closed again and the system restarted. Both were returned to 31˚C.

In addition, an additional donor cornea was infected in the ASM with the same protocol and observed with an ophthalmologic slit lamp after fluorescein staining [26] (Fluorescein Faure single dose, SERB Laboratories, Paris, France) to show clinician readers the similarity in pattern to dendritic keratitis *in vivo*.

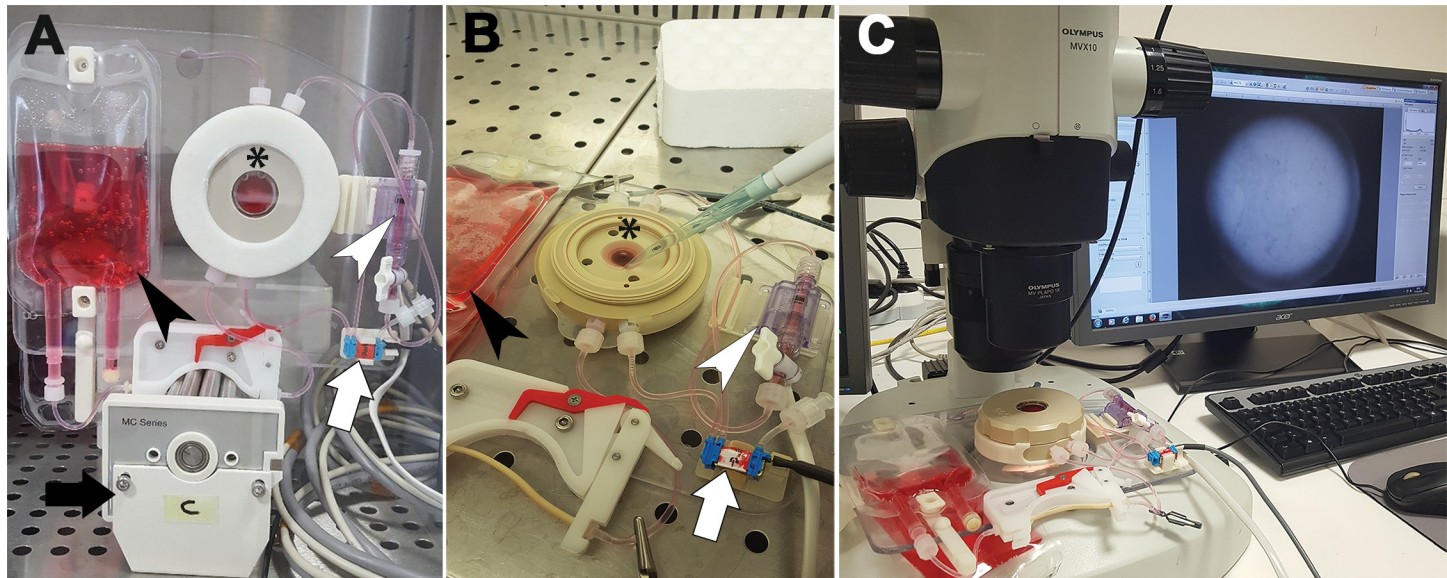

**Fig 1. Steps of herpetic infection and monitoring in the active storage machine (ASM).** (A) Two-week epithelial restoration step. (B) Inoculation with HSV-1 suspension. The ASM epithelial lid was opened under a laminar flow hood. (C) Daily monitoring of the ASM using a macro zoom microscope. Asterisk: corneal support machined in polyether ether ketone; black arrowhead: storage medium tank; white arrowhead: pressure sensor; black arrow: peristaltic pump; white arrow: solenoid valve.

The pairs of corneas used as uninfected controls in the ASM and OC groups underwent the same treatment as the infected corneas, except for exposure to the virus.

## Monitoring of herpetic infection

After infection by HSV-1, corneas were monitored daily for 3 to 6 days (depending on the rate of lesion development) by imaging the epithelium under a macro zoom microscope with retro illumination (MVX-10 macroscope run by the CellSens Dimension imaging software, Olympus, Tokyo, Japan) (Fig 1C). A x1 objective allowed single-field observation of the whole human cornea. To compensate for the dome shape of the cornea, a z-stack of six images spaced 600 μm apart, from the limbus to the corneal center, was acquired in bright field. Special interest areas were also observed with a x6.3 objective, using a z-stack of images spaced 85 μm apart. The in-focus images of the epithelium were then reconstructed using the 3-D Extended Depth of Field plugin (http://bigwww.epfl.ch/demo/edf, in the public domain) in ImageJ (National Institutes of Health, Bethesda, MD, USA).

## Immunostaining on living tissues

As one of our secondary objectives was to study the possibility of *in vivo* immunolabeling, at the end of the experiment immunolabeling was performed on unfixed tissue. Corneas were gently rinsed in PBS and incubated at 37˚C for 30 minutes with a mouse monoclonal anti-HSV1/2 glycoprotein B (gB) primary antibody (Sigma, Saint-Quentin-Fallavier, France), diluted 1:300 in PBS. The secondary antibody was Alexa Fluor 488 goat anti-IgG2B (Invitrogen, Eugene, OR, USA), diluted 1:500 in PBS and incubated for 30 minutes at 37˚C. Lastly, nuclei were counterstained with Hoechst 33342 (8μM; Sigma) for 5 minutes at room temperature (RT). After each step, corneas were triple-rinsed in PBS. A z-stack of tagged image format file (TIFF) images was acquired as previously described using the same fluorescent macroscope with the x1 and x6.3 objectives. Images were acquired in bright field and in fluorescence mode with filters for fluorescein isothiocyanate (FITC, ex: 450–490/ em: 500–550 nm) for anti-gB antibody and for Hoechst (ex: 300-400/em: 410–510 nm). As with monitoring of the infection, the in-focus images of the epithelium were reconstructed for each channel. In the ASM group, all immunolabeling steps were done on the cornea secured inside the ASM. The lid was opened and the epithelial chamber was used as a reservoir for the successive reagents. In the OC group, incubations and rinses were done in a concave plastic holder. The same immunostaining protocol was applied to uninfected control corneas.

After immunolabeling, corneas were removed from the ASM or the OC vial, rinsed in PBS and cut into seven wedge-shaped parts for immunohistochemistry (one part), immunofluorescence (four parts: three for flat mounting, one for cross-section), transmission electronic microscopy (one part) and PCR (one part).

## Immunostaining on flat-mounted corneas

To compare immunolabeling on living tissue with standard immunohistochemistry, we fixed the corneas and processed them with protocols that we previously optimized for whole mounts [27, 28]. Briefly, three wedge-shaped parts (or the whole cornea for nerve mapping) were fixed immediately for 45 minutes either in 0.5% paraformaldehyde (PFA) in PBS pH 7.45 at RT or in methanol at RT, depending on the antibody. After fixation in PFA, cell membranes were permeabilized using 1% Triton x-100 in PBS for 10 minutes at RT. The non-specific binding sites were then blocked by incubation for 30 minutes at 37˚C with blocking buffer based on PBS supplemented with 2% heat-inactivated goat serum and 2% bovine serum albumin (BSA, Thermo Fisher Scientific, Waltham, MA, USA). All primary antibodies listed in S1 Table were

diluted 1:300 in the blocking buffer. gB/E-Cadherin, gB/Cytokeratin 3 (K3), gB/Desmocollin 2/3 (DSC2/3) co-staining were systematically performed, except for nerve mapping on whole cornea, where only the anti-Neurofilament-L (NF-L) antibody was applied. The cornea wedges were fully immersed in this solution and incubated overnight at 4˚C. Secondary antibodies were Alexa Fluor 488 goat anti-mouse IgG2B, Alexa Fluor 555 goat anti-mouse IgG1 and Alexa Fluor 555 goat anti-rabbit IgG (respectively A21141, A21127 and A32727, Invitrogen). Incubation with secondary antibodies diluted 1:500 in blocking buffer was done for 1 hour at 37˚C (or for 2 hours at 37˚C for nerve mapping of the whole cornea). Finally, nuclei were counterstained with 2μM TO-PRO-3 Iodide (T3605, Molecular Probes, Invitrogen) in PBS for 5 minutes at RT, except on the whole cornea where the nuclei were counterstained with DAPI at 10μg/mL in PBS for 1 hour at RT. After each step, the wedges were tripled-rinsed in PBS, except between blocking of non-specific protein binding sites and incubation with the primary antibody. Gentle agitation was necessary throughout the process. The corneal wedge was then placed on a glass slide, covered with Vectashield medium (Vector Laboratories, Burlingame, CA, USA), and gently flattened using a large glass coverslip held by adhesive tape. Experiments were done at RT unless otherwise stated. Images were captured with a laser scanning confocal microscope (FV1200; Olympus) equipped with the FV10-ASW4.1 imaging software. Four lasers were available: 405nm, 473nm, 546nm and 635nm; and the emission filters were respectively 430-470nm, 490-525nm, 560-620nm and 655-755nm. A large surface was mosaicked by Multi Image Alignment in X, Y and Z. The uninfected control corneas underwent the same protocol.

## Histology and immunostaining on cross-sections

One wedge of corneas was fixed in 4% PFA for 24 hours at RT, dehydrated through ascending concentrations of ethanol, and embedded in paraffin. Cross-sections 7μm thick were cut, rehydrated, and stained with hematoxylin, eosin, and saffron. Bright-field TIFF images of the cross-sections were acquired using an inverted microscope (IX81; Olympus). Another wedge of corneas was embedded in optimal cutting compound (CellPath, Newtown, UK) and frozen using 2-methylbutane (Sigma) and liquid nitrogen. Samples were stored at -20˚C. Tissue sections (10μm thick) were cut using Cryostat Leica CM1950 (Thermo Fisher Scientific) and spread on Surgipath X-tra Adhesive slides (Leica Biosystems, Nussloch, Germany). Nonspecific binding sites were blocked by incubation for 30 minutes at 37˚C with blocking buffer, based on PBS supplemented with 2% heat-inactivated goat serum (Eurobio) and 2% bovine serum albumin (Thermo Fisher Scientific). Slides were incubated at 37˚C for 1 hour with the primary antibody. Primary antibodies were supplied as listed in S1 Table. Nonspecific rabbit and mouse immunoglobulin G (IgG; Zymed, Carlsbad, CA, USA) were used as primary antibodies for negative controls. These controls were performed for each cornea. Secondary antibodies were the same as listed above. Finally, the slides were mounted using Vectashield medium. Images were captured with a confocal microscope (IX83 Fluoview FV-1000, Olympus), equipped with the Olympus Fluoview software. The control corneas underwent the same protocol. Percentages of infected stromal surface in ASM and OC were described as a mean +/- standard deviation and compared using non-parametric paired Wilcoxon signed rank test with a 5% significant level.

## Transmission electronic microscopy

One wedge of corneas was fixed in 1% glutaraldehyde/0.5% PFA (diluted in 0.1 M Na/diK mono buffer at pH 7.4). It was then post-fixed in 1% osmium tetroxide (diluted in 0.1 M cacodylate buffer) for 1 hour, dehydrated through ascending concentrations of ethanol and integrated into EPON resin. Ultrafine sections (90 nm) were stained with uranyl acetate and lead

citrate. Observations were made with a transmission electron microscope (H-800; Hitachi, Tokyo, Japan) equipped with a CCD camera (XR40; AMT, Danvers, Massachusetts, USA).

## Polymerase chain reaction

Virological and molecular methods were used to investigate the corneas and detect whether HSV-1 DNA was positive in the samples. Briefly, one wedge of the corneas was fixed immediately for 45 minutes in methanol at RT and then placed in PBS pH 7.45 at 4°C until processing. Viral DNA was extracted from each cornea wedge using the high pure viral nucleic acid kit (11858874001, Roche LifeScience) with some adaptations described in the supporting information (S1 Fig). PCR was then carried out on the purified viral nucleic acids from each sample using the HSV1 HSV2 VZV R-gene real time detection and quantification kit (69-004B, bio-Mérieux, Marcy l'Etoile, France). To estimate the number of cells in a sample and compare the viral load (expressed as number of HSV-1 genome copies/1000cells) of each sample, a cellular control for Real-Time PCR amplifying a part of the housekeeping gene HPRT1 encoding for Hypoxanthine-guanine phosphoribosyltransferase 1 was used (CELL Control r-gene®, 71–106, bioMérieux) as previously described [29]. All PCRs were performed in duplicate per the supplier's recommendations and using a real-time PCR system (7500, Thermo Fisher Scientific). The average of the two measurements was used.

## Statistical analyses

Data were represented by scatter plots and compared using non-parametric tests (Mann-Whitney test and paired Wilcoxon signed rank test) with a 5% significant level.

## Results

### Dendritic lesion patterns developed in the ASM look like clinical patterns

After fluorescein instillation, the epithelial lesions observed in the ASM 4 days after inoculation formed a dendrite similar to that of a common epithelial HK (Fig 2). To our knowledge, dendritic lesions have not previously been reported in human cornea *ex vivo*.

### Only ASM-stored corneas developed dendritic or geographic epithelial ulcers

In the ASM group (n = 6), dendritic (three cases) or geographic ulcers (two cases) typical of HK appeared on the infected corneas at day 3 or 4. The course of the infection was observed for 5 to 7 days. Macroscopic patterns (dendrites and/or geographic ulcer) all started from the initial mechanical linear central ulcer. Other small ulcers, observed at the periphery of the cornea, were not necessarily related to the primary structure (Fig 3A). In one case, no ulcer was observed during the 7-day follow-up and the pre-infection scratch actually healed at D2. Notably, this cornea had the longest OC time (51 days) before the ASM rehabilitation phase (S2 Table). The geographic or dendritic patterns obtained on each of the six corneas are illustrated in the supporting information (S2 Fig).

In the OC group (n = 6): in two corneas (discarded by an eye bank), no specific HSV infection pattern was observed. When the central scratch was made, the epithelium appeared extremely thin, probably with one layer, partially abraded, and cells desquamated rapidly after infection. In the other corneas (one from eye banks, three from bodies donated to science) the epithelium appeared slightly thicker, probably comprising 2–3 cell layers. The initial scratch was clearly visible in the first few days. At D3 or D4, some buds appeared around the scratch, delimiting an infected area. At D6 or D7, most central cells had desquamated, leaving the bare

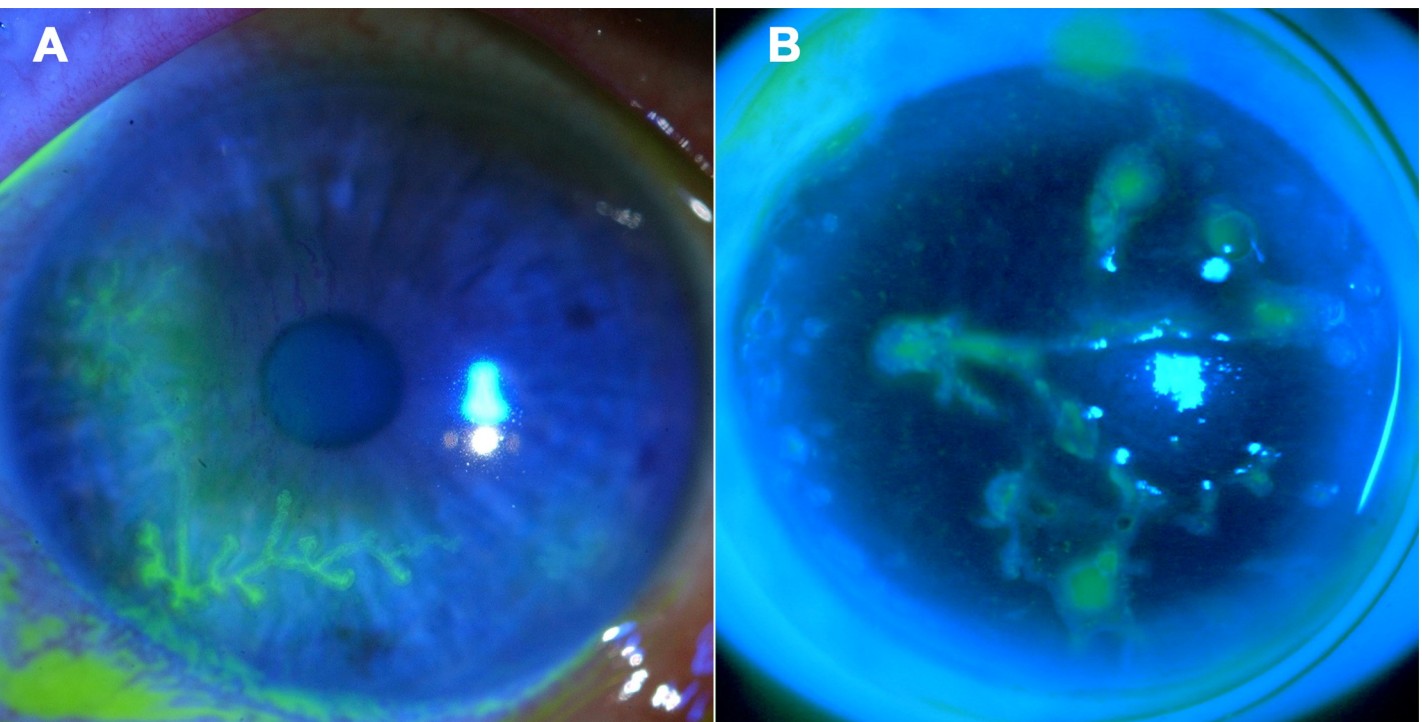

**Fig 2.** Observation with a slit lamp and cobalt blue filter of dendritic herpetic keratitis in a patient (A, personal database of GT) and of a dendritic epithelial lesion obtained in the ASM at day 4 post infection (B).

Bowman's layer and forming a large rounded ulcer (Fig 3A and S2 Table). Regardless of group, no donor-age effect was observed in the experiment findings.

### Immunostaining of living (unfixed) tissues reveals infected epithelial cells

In the ASM group: in all corneas, immunolabeling revealed infected epithelial cells. The cornea without a macroscopic pattern presented only a few infected cells in the periphery. Nuclear counterstaining allowed perfect visualization of the edges of the main ulcers (dendritic or geographic) and also of small satellite ulcers nearer the periphery. At the bottom of the ulcers, the keratocyte nuclei were clearly visible through transparency, indicating that the entire epithelium had desquamed in these areas. The HSV-gB was expressed in epithelial cells on the edges of the ulcers. Staining was located at membrane level. Positive cells were mostly rounded, corresponding to the CPE of HSV. Outside the herpetic lesions, the epithelium remained multilayered (Fig 3B).

In the OC group, two corneas had no epithelial cells left and thus no epithelial labeling. In three other corneas, the epithelial layers were highly abraded and appeared to comprise a single cell layer. This layer was strongly labeled with anti-gB antibody, but with lower fluorescence intensity than in the ASM group. The cells were not necessarily rounded. In one cornea only, a geographic ulcer surrounded by a few peripheral satellite ulcers and, slightly further away, by multilayered epithelium was observed. Cells expressed HSV-gB on the edges of the ulcers, as observed in the ASM group. The uninfected controls were unstained (Fig 3B).

### The ASM preserves epithelial structures better than conventional organ culture

Uninfected controls in the ASM or OC had a stratified corneal epithelium of 3–5 layers. Epithelial cells and keratocytes were unstained by the anti-gB antibody, and the nuclei were

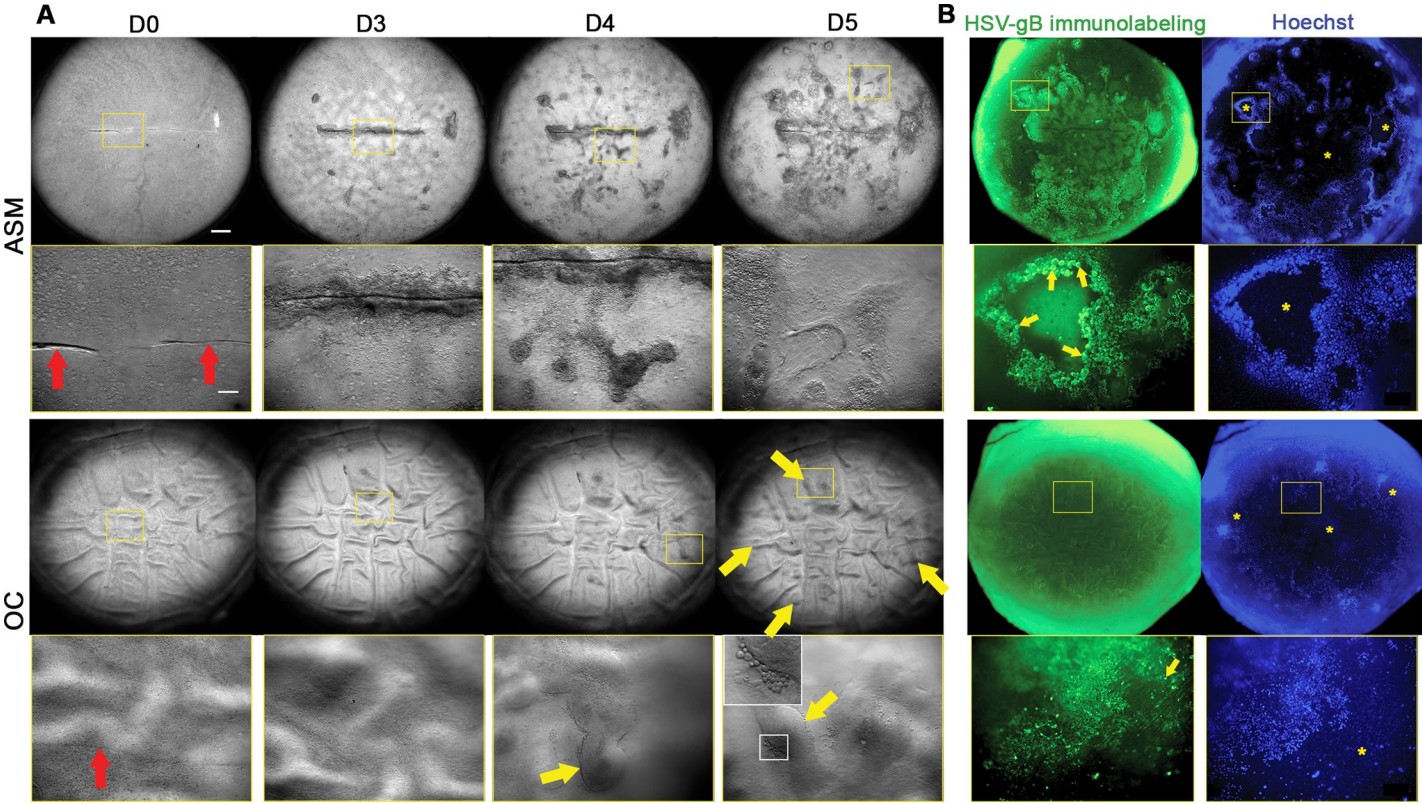

**Fig 3. *Ex vivo* models of herpetic lesions on corneas.** (A) Chronology of the development of herpetic lesions on the corneas stored in the active storage machine (ASM) versus organ culture (OC). Bright field macroscopic observation (x1 and x6.3 zooms) after scratching (D0) and infection at D3, D4 and D5. In the OC cornea, the red arrows show the initial scratch; and the yellow arrows, the "buds" that appeared at the edges of the infected area. Note that the OC cornea presented a characteristic tessellated pattern (highlighted by retro illumination) corresponding to the numerous deep posterior folds caused by stromal edema. On the contrary, the ASM-stored cornea was thinner, presented no folds and thus appeared smooth. (B) Anti HSV-gB immunolabeling on unfixed whole human corneas. Observation with a fluorescence macro-zoom microscope (x1 and x6.3 zooms) of the corneal epithelium infected with HSV-1 in the ASM or OC and labeled with anti-HSV-gB and Hoechst after 5 days' incubation in the ASM or OC. Yellow asterisks show full-thickness epithelial ulcers where nuclei of Hoechst-positive keratocytes were visible through transparency. Yellow arrows show the rounded infected cells demonstrating the cytopathogenic effect (CPE). Note that the corneal limbus and the scleral rim were strongly autofluorescent on this wavelength, but this did not prevent corneal imaging. Scale bar: 1000 μm for low magnification images and 200 μm for high magnification images.

normal. In the ASM or OC, cells had a normal staining pattern for the three epithelial markers: superficial cells expressed K3 in their cytoplasm, the plasmic membrane of superficial and intermediate cells expressed DSC2/3, and the plasmic membrane of all cells expressed E-cadherin, lying on a epithelial basement membrane positive for laminin-5 (Figs 4 and 5).

For infected corneas, in the ASM group, corneal epithelium was stratified with 2–6 largely disorganized layers (except for one cornea with only one cell layer). The CPE was clearly visible in hematoxylin, eosin and saffron staining and in immunohistochemistry (Fig 5). Infected epithelial cells were identified on all six corneas. gB+ infected cells demonstrating a CPE were not or very weakly labeled with E-cadherin, K3 or DSC2/3, contrary to gB+ cells that retained their shape and slightly co-expressed E-cadherin, K3 and DSC2/3. A clear CPE was also confirmed by nuclei of gB+ cells that were always larger than in normal cells, leading to a higher nuclear-cytoplasmic ratio due to nuclear replication of HSV-1. The CPE observed in corneal epithelial cells is similar to that observed in HSV-1 infected fibroblast cell culture [31].

In the OC group, the epithelium was absent in two corneas. The other corneas had an epithelium of 1–4 largely disorganized layers, and infected gB+ epithelial cells were identified.

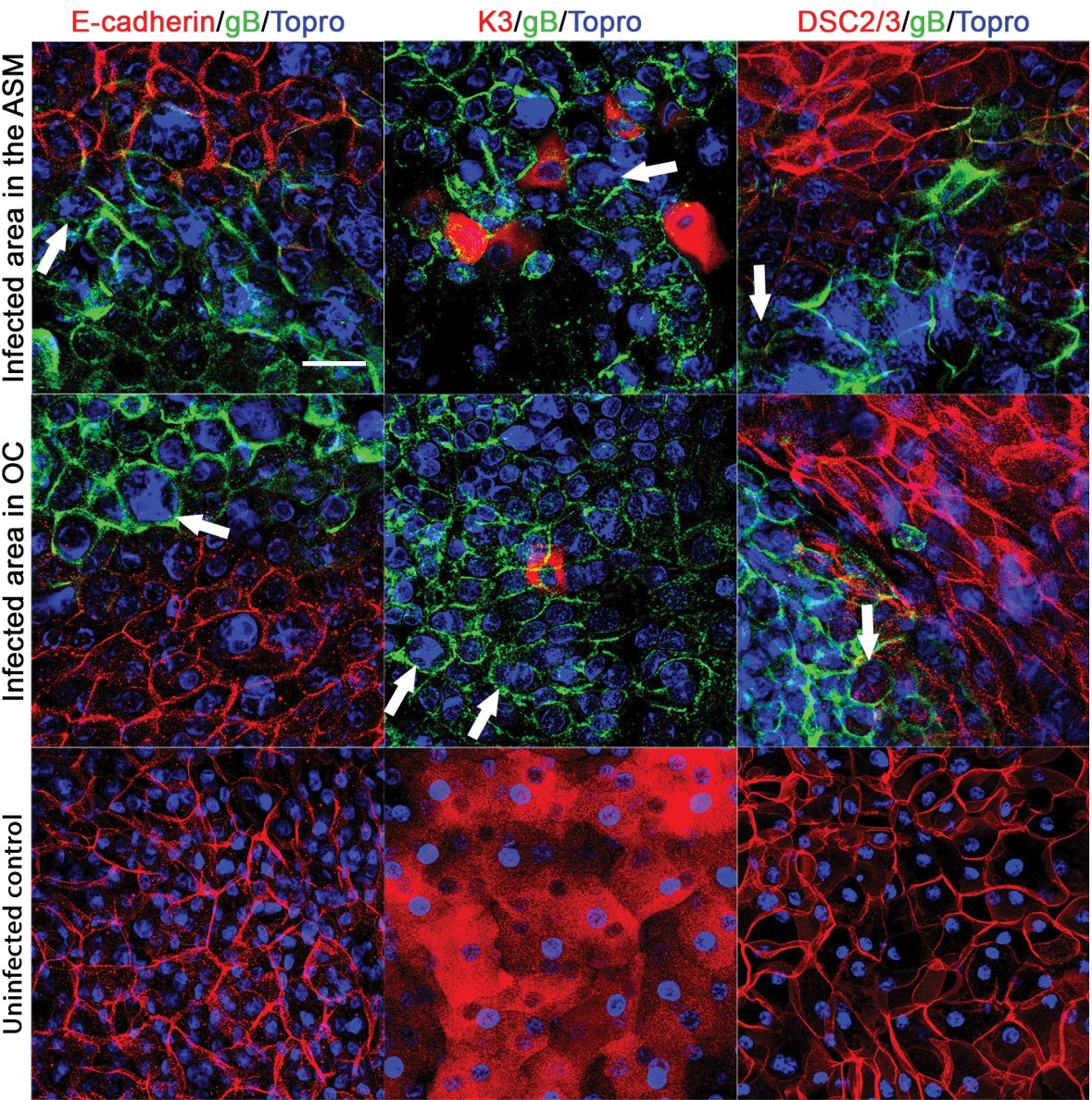

**Fig 4. En face view of the corneal epithelium after HSV-1 infection.** Confocal microscopy of corneal epithelium infected in the active storage machine or in organ culture and labelled with anti-gB/E-cadherin after methanol fixation or K3 or DSC2/3 after 0.5% PFA fixation, a mean 5 days after HSV-1 inoculation. The white arrows show the rounded infected cells exhibiting the HSV-1 induced cytopathogenic effect with a higher nuclear-cytoplasmic ratio. No gB staining was observed in negative controls. Nuclei were stained with To-PRO 3 Iodide. Scale bar: 50 μm.

Similarly to the ASM group, areas of infected cells were not or only weakly labeled by k3, E-cadherin and DSC2/3 (Figs 4 and 5).

In addition, infected keratocytes were observed in five out of six corneas in the ASM group and in five out of six corneas in the OC group, as far as 98 μm and 172 μm respectively beneath Bowman's layer. On the cross-sections, the percentage of stromal surface showing infected keratocytes was 1.7 +/-1.2% in the ASM versus 10.3+/-12.9% in OC (P = 0.687 non parametric paired Wilcoxon signed rank test).

With transmission electron microscopy, in the ASM group, in five of the six corneas, the epithelium was mature and multilayered with microvilli in the most superficial cells. Several

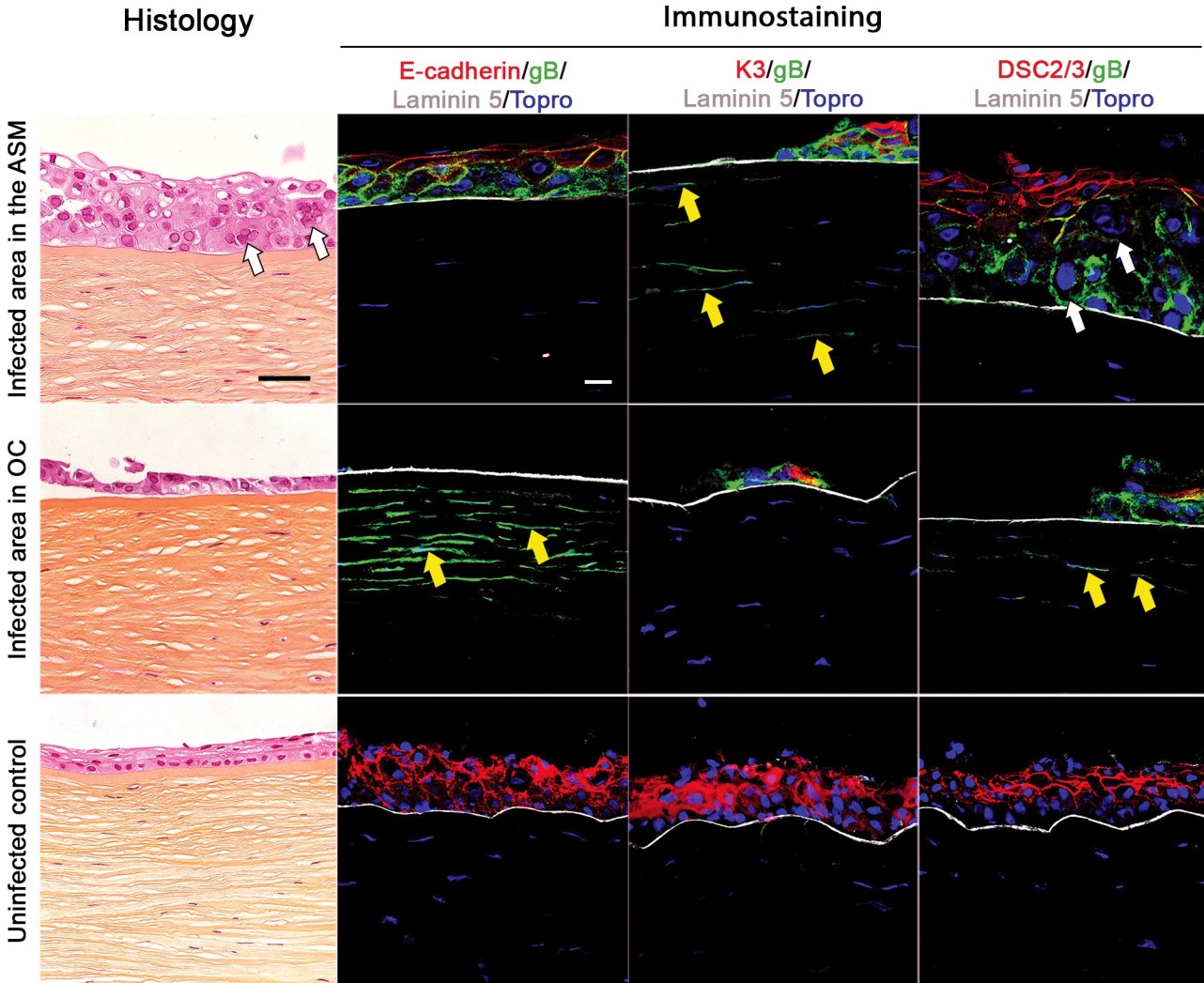

**Fig 5. Cross-section of the corneal epithelium and anterior stroma after HSV-1 infection.** Only infected areas are shown. Histology (hematoxylin, eosin and saffron, scale bar: 50 μm) and immunohistochemistry of paired corneas infected in the active storage machine (ASM) or in organ culture (OC) (scale bar: 20μm) were compared with uninfected corneas (the ASM control group is presented here). Cells infected by HSV-1 expressed HSV-1 gB. Epithelial organization and maturation were labeled by the three epithelial markers (K3, E-Cadherin, DSC2/3), and the epithelial basement membrane integrity by expression of laminin-5. Yellow arrows mark gB+ keratocytes and white arrows marked the cytopathogenic effect (CPE) in epithelial cells. No gB staining was observed in uninfected controls. The thickness of the corneal epithelium in the ASM was greater than in passive OC because the ASM can regenerate an epithelium with more layers than in passive OC [30] and, in this thicker epithelium, the CPE further increased the difference between the ASM and OC. Nuclei were stained with To-PRO 3 Iodide.

signs of active viral replication were observed (Fig 6): i/numerous enveloped viral particles in the intercellular spaces (with disorganized tight-junctions and desmosomes) or in the cytoplasm of the epithelial cells (sometimes in vacuoles) ii/viral capsids in the nuclei (Fig 6A and 6B); iii/ enveloped viral particles were also observed between Bowman's layer and the basal epithelial cells. Interestingly, in the only cornea that did not present a typical ulcer, viral capsids were only observed in nuclei, and no enveloped viral particle was observed in the intercellular spaces. In addition, epithelial cells were tightly packed, with clearly visible intact tight-junctions and desmosomes (Fig 6C). In one cornea, the epithelium could not be observed, but enveloped viral particles were observed on Bowman's layer. In all corneas, we were unable to observe infected keratocytes in the anterior stroma.

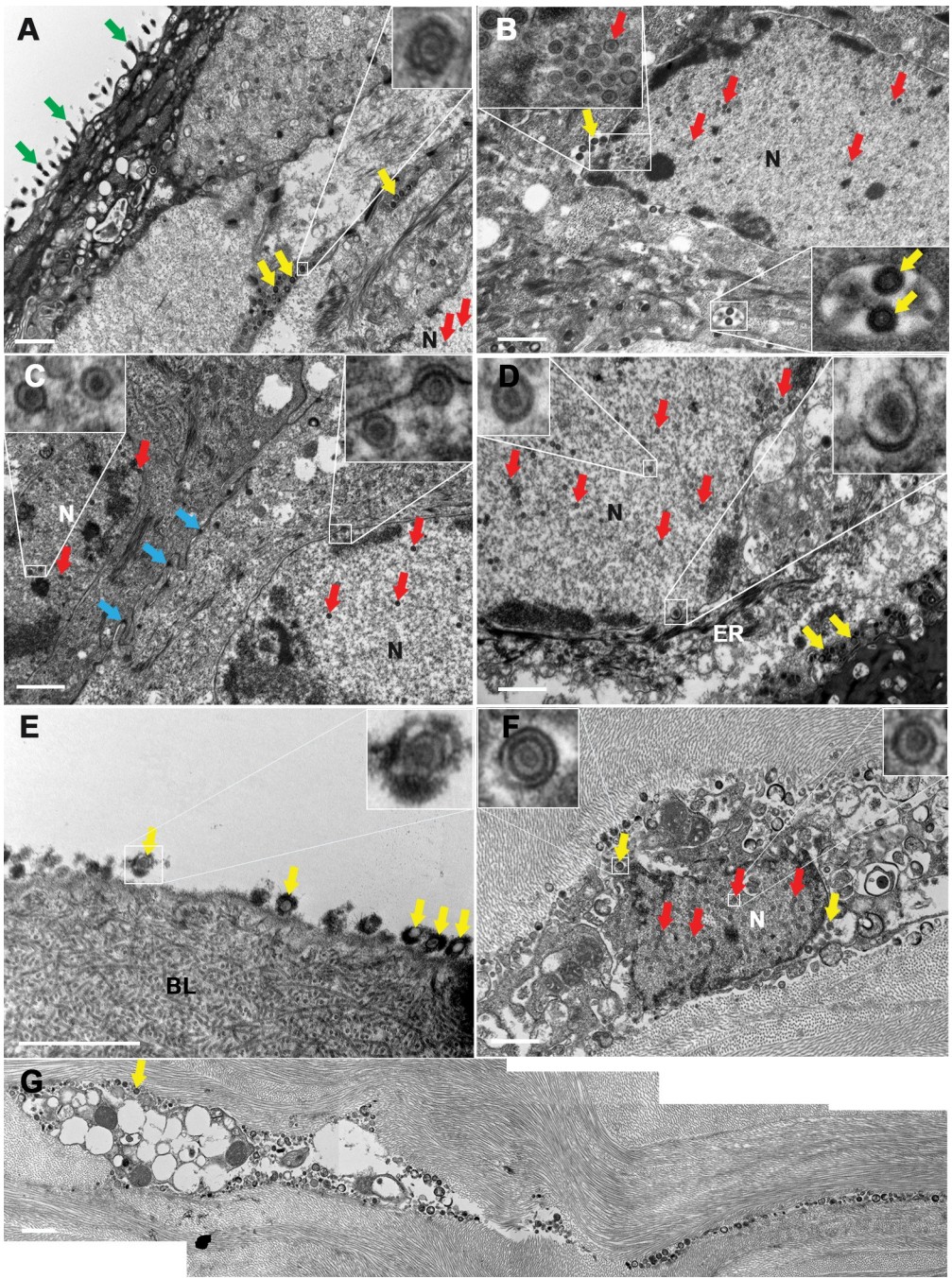

**Fig 6. Ultrastructure of the corneal epithelium after HSV-1 infection.** Corneas were stored either in the active storage machine (ASM) (A to C) or in organ culture (D to G). The microvilli (green arrows) present only on the superficial cells of corneas in the ASM demonstrated normal epithelial maturation. (A and B) Corneas with dendritic or geographic ulcers. Enveloped viral particles in the intercellular spaces or in cytoplasmic vacuoles. Viral capsids in the nucleus. (C) Cornea without macroscopic lesion. Viral capsids in the nuclei of epithelial cells. Desmosomes were intact (blue arrows). (D) Enveloped viral particles in the intercellular spaces and viral capsids in the nuclei of epithelial cells. (E) Enveloped viral particles on Bowman's layer in the absence of epithelial cells. (F) Infected keratocyte with viral capsids in the nucleus and enveloped viral particles in the disorganized cytoplasm. (G) Reconstruction of an infected keratocyte. BL: Bowman's layer, ER: Endoplasmic reticulum, N: Nucleus. Yellow arrow: enveloped viral particle. Red arrow: Viral capsid. Scale bar: 2 μm.

In the OC group, signs of viral replication were also observed (Fig 6D). In the three corneas without epithelium, enveloped viral particles were observed on Bowman's layer in two cases (Fig 6E). In one of these cases, enveloped viral particles were also observed in the completely disorganized cytoplasm of keratocytes (Fig 6F and 6G). Similarly to the ASM group, in the three corneas with epithelium, viral capsids in the nuclei and enveloped viral particles in the intercellular spaces were observed.

## All intraepithelial nerve structures disappeared after cornea storage

Corneal nerve mapping demonstrated that the intraepithelial nerves (subbasal plexus and intraepithelial nerve endings) disappeared after 2 weeks in OC followed by 2 weeks in the ASM (Fig 7).

## No difference in viral load between the ASM and organ culture

Viral DNA of HSV-1 was detected in all samples but the number of copies in uninfected controls was almost negligible. A significant difference in viral load between infected and uninfected samples was observed in the ASM group (respectively 21254 +/- 17588 vs 0.9042 +/- 0.8355 genome copies/1000cells) and the OC group (respectively 21419 +/- 14091 vs 0.1351 +/- 0.1435 genome copies/1000cells) (both p = 0.0238, non-parametric Mann-Whitney test). No significant difference was detected between the infected samples in the ASM and in OC (p = 0.4375, non-parametric paired Wilcoxon signed rank test).

## Discussion

Until now, the feasibility of *ex vivo* HK models in human cornea was limited for several reasons: 1/ corneal donation is generally far below demand, be it for transplantation [32] or research [33]; 2/ Available corneas are most often declared unsuitable for transplantation by eye banks and, therefore, have already been stored for several days or weeks; 3/ Corneal epithelium quickly degrades after donor death (drying out, abrasion); 4/ Its quality rapidly deteriorates during storage in hypothermia [34] and in OC [35]; 5/ the CPE characteristic of herpes infection causes further deterioration of the already impaired epithelium (as confirmed by our OC group).

By restoring corneas to a much more physiological environment than passive OC, our ASM rehabilitates corneas discarded by eye banks, which have a severely degraded epithelium. By regenerating a multilayered and mature epithelium, very close to its normal state [30], and maintaining it over time, the ASM makes corneas suitable again for use in certain models, such as the one presented here.

We already demonstrated that the ASM improved endothelial survival and epithelial maturation in the short term (7 days) in porcine corneas [25] and in the long term (28 days and 3 months) in human corneas [23, 24]. Importantly, in both these previous studies, animal and human corneas were fresh and had only been stored in the ASM since procurement. Here, we used the ASM to rehabilitate corneas already stored in OC for several days or weeks. And we confirm that the mature epithelium regenerated on human cornea allows faithful reproduction of both dendritic and geographic HK patterns. Interestingly, the only exception (absence of dendrite) was the cornea that underwent particularly long OC (51 days) before placement in the ASM. Despite the regeneration of a multilayered epithelium and a proven viral infection, no dendrite or ulcer was observed. This suggests that corneal damage caused by very long OC does not allow complete tissue regeneration. This must be confirmed by further studies. In addition, the absence of a difference in viral load between OC and the ASM suggests that, although the number of epithelial cells in the ASM is much higher than in OC because the

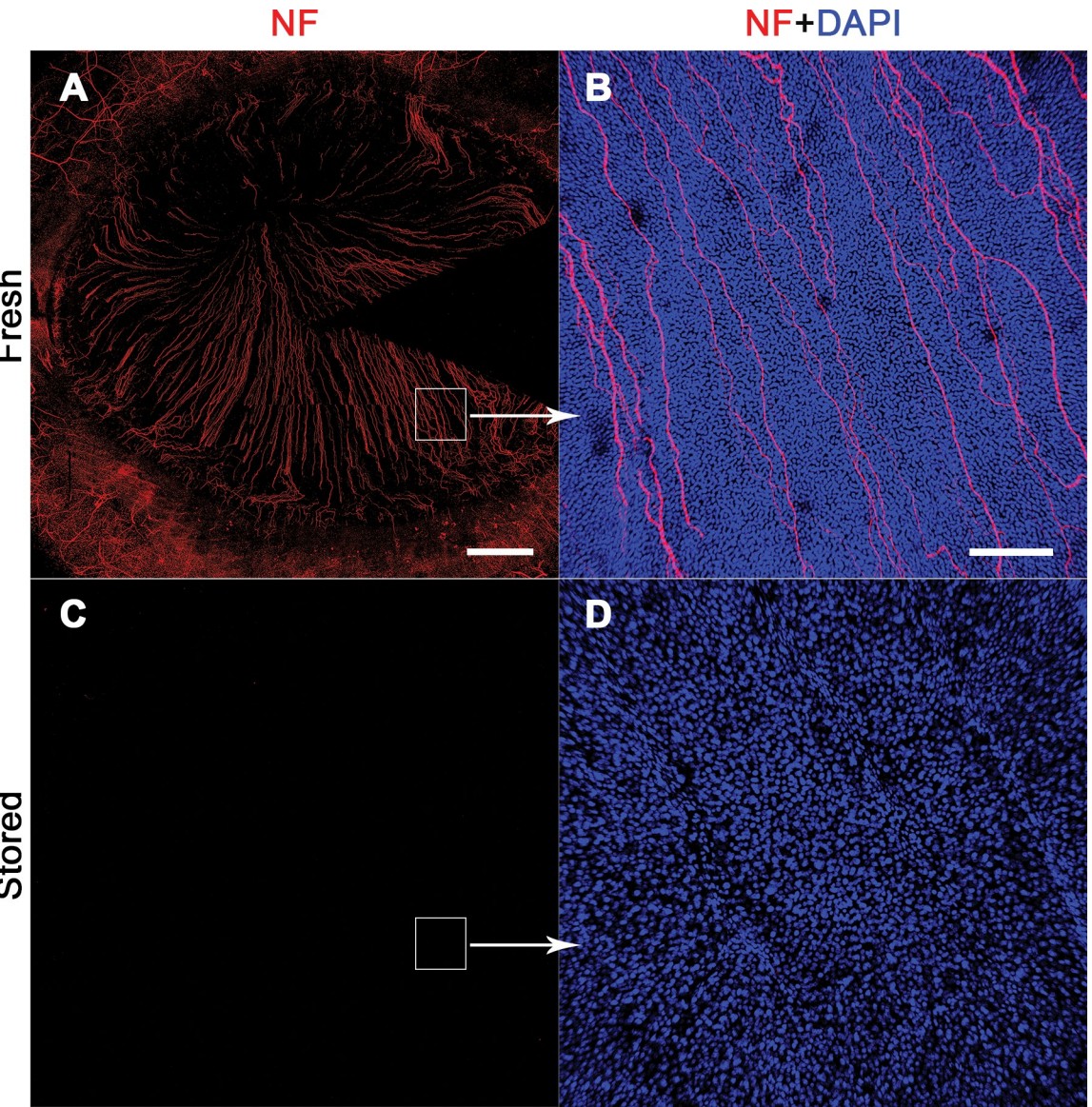

**Fig 7. Disappearance of intraepithelial nerve structures after organ culture and storage in the active storage machine.** Nerves were revealed by immunostaining of the Neurofilament-L (NF in red). One cornea was studied fresh, immediately after retrieval (A) and its paired cornea was studied after 14 days' organ culture storage and 14 days' additional storage in the active storage machine (C). (A and C) Nerve map of the whole flat-mounted cornea with its limbus and proximal conjunctiva. Scale bar: 2 mm. (B and D). Close-up of the nerves, and of epithelial cell nuclei labeled with DAPI (in blue) (x10 objective). The epithelium of the stored cornea was very disorganized and had far fewer cells. Scale bar: 200 μm.

epithelium is multilayered, the number of infected cells is similar in both groups after the same starting inoculum and the same incubation period. Moreover, the far heavier gB+ staining of keratocytes in OC may be because, in OC, the virus reaches Bowman's layer more quickly, and can cross it. This model thus reproduces the infectious phase of herpetic stromal keratitis, as it progresses through the corneal layers. This raises the possibiity of adding inflammation mediators, and even immune cells, to the ASM to produce a *ex vivo* model of immune stromal keratitis.

Furthermore, the ASM will likely improve epithelial-stromal-endothelial interactions. We observed that keratocyte infection appeared less widespread in the ASM than in OC (though

not reaching the level of statistical significance). This is an additional argument for the greater integrity of the ASM's epithelial barrier. Finally, the design of the ASM as a hermetic system including two observation windows allows infection monitoring with no risk of contamination.

There are various models for studying ocular herpes viruses, ranging from *in vitro* mono-layer cell culture over *ex vivo* stored corneas to *in vivo* eye infection models, most commonly with rabbits and mice [16]. Each model has some limitations but *ex vivo* models are more physiologically relevant than cell culture systems, and also less expensive and more ethical than *in vivo* models. E*x vivo* models can be used to evaluate new treatments [14, 21] and poten-tial diagnosis procedures [18], and to study the physiopathology of the disease [15, 17, 20, 22].

In our protocol, the small epithelial lesion made at D0 does not correspond to the patho-physiology of epithelial HK, in which the virus reaches the corneal epithelium via nerve end-ings. However, this step is necessary because the ASM restores a normal epithelial barrier, and also because of the viral inoculation method and of the rapid degeneration of intraepithelial nerve endings. The development of lesions from the central scarification shows it to be the main entry point for the virus. Its initial intracellular penetration must thus be carried out on basal or intermediate epithelial cells, which are metabolically more active. This scarification was already used in several *ex vivo* human [15, 18] and porcine [21, 22] models and also in *in vivo* mice and rabbit models [36–40], in particular to promote infection on a healthy cornea, where the virus is quickly eliminated by eyelid blinking.

The various *ex vivo* models use corneas, scarified or not prior to infection, infected by direct contact on the corneal epithelium of the viral strain containing between $1x10^4$ and $5x10^6$ PFU for 1–24 hours [14, 15, 17–22]. For instance, Novitskaya *et al.* directly incubated the human corneas (fresh, we assume) in the medium containing the viral strain for 3 hours at 37˚C, fol-lowed by 21–69 hours of incubation in a fresh sterile medium. Some corneas had no epithe-lium left while others had a disorganized superficial epithelium, but no dendritic pattern or geographic ulcer was described [18]. Alekseev *et al.* described another *ex vivo* model that involved placing human and rabbit corneas on agar, epithelium side down, into the virus-con-taining medium for 1 hour followed by incubation in a culture medium, epithelium side up [19]. This model has since been used in several studies [14, 17, 21]. Based on this model, an air-liquid culture model was also described to study acute ocular herpesvirus infection using canine corneas [20]. Lastly, Thakkar *et al.* [22] showed that a dendritic pattern could be obtained on fresh porcine corneas 96 hours after inoculation. Corneas with small epithelial needle punctures were immersed epithelium side up in the viral solution at $1x10^5$ to $5x10^6$ PFU for 24 hours, then further cultured in a fresh sterile medium. We believe that the dendritic lesions only developed thanks to the excellent quality of the fresh epithelium of the eyeballs, used before post-mortem or storage-induced lesions occurred.

The mechanisms behind the formation of typical dendritic or geographic HSV lesions are not fully understood. Further, punctate corneal vesicles in the epithelium are one of the first manifestations of acute HSV infection, and coalesce into branching dendritic lesions. Geo-graphic corneal ulcer may progress from this stage and contain live virus on the edges of the ulcer [41, 42]. The epithelial lesions are caused by the herpetic CPE (replication followed by cell destruction) and the immune system and/or the many intraepithelial nerves are suspected of roles in forming these pathognomonic patterns. Our model shows that neither of these two elements is required. We are the first to demonstrate that the intraepithelial nervous plexus is unnecessary for the development of dendritic or geographic ulcers in humans after reactiva-tion of the virus *in vivo*. There seems to be only one reference, dating from 1970, which also formulated this hypothesis but on a very different model. The authors used a model of pene-trating keratoplasty in rabbits with scraping of the donor epithelium [39]. In this model the

peripheral nerves persist and, as the authors point out, regeneration of the intraepithelial nerves is possible within a few days, which is not true of our model.

The absence of immune response effectors and of functional innervation are the two main limitations of *ex vivo* models, including ours. Indeed, concerning the immunological aspect, OC-stored corneas quickly lose some of these passenger leukocytes (in particular the antigen-presenting dendritic cells) [43], which have been deemed to help reduce the risk of graft rejection in patients [44] and in mice [45]. Moreover, *ex vivo* models do not generally allow immune cell influx (the ASM is no exception). Consequently, the model described here is of no use for evaluating anti-inflammatory or immunomodulatory interventions in HK. But the absence of any host response in our model did not prevent the formation of pathognomonic HSV ulcers, although we used corneas with a prolonged OC time. Therefore, the absence of development of dendritic lesions in other *ex vivo* models [14, 15, 17, 19–21] is most likely due not to the absence of infiltration by immune cells but to the deterioration in quality of the epithelium itself in these first-generation models. However, concerning the neurological aspect, the subbasal nerve plexus rapidly degenerates post mortem or during OC [46]. As demonstrated in this paper, the corneas rehabilitated in the ASM therefore have no intraepithelial nerves. *In vivo*, epithelial HK is due not to primary infection but to recurrence from an established latency [2]: the virus reaches the corneal epithelium from the trigeminal ganglia via the nerve endings and infects the first epithelial cells (probably a small number). Our study using the ASM can help better understand the genesis of dendritic and geographic patterns, even if the infection is induced by direct contact.

Although we succeeded in reproducing epithelial HK, thus demonstrating the absence of immune-system involvement in its formation, our model is obviously unsuited to reproducing stromal HK, in which the role of immunity is established [47]. However, given the possibility of long-term corneal storage in the ASM, we can envisage developing more complex models in which certain immune mediators (chemokines, cytokines) or actors (Langerhans antigen-presenting cells, neutrophils and lymphocytes) could be injected into the model in sequential pairs. We could thus study more precisely the interactions between the virus and the immune system. Based on these new models, it will also be possible to test new drugs (antiviral agents, anti-inflammatory or immunomodulatory drugs).

## Conclusion

Our model, with corneas rehabilitated in an ASM, shows that the formation of the dendritic or geographic pattern characteristic of epithelial HK only requires the presence of a mature multilayer epithelium. The intraepithelial nerve plexus and immune cells are unnecessary. *In vivo*, the nerves enable the virus to reach the epithelium but do not subsequently guide cell damage. Virus-epithelial cell interaction alone creates one of the two patterns. This innovative model of epithelial HK on human cornea in the ASM could be very useful in both basic and therapeutic research.

## Supporting information

**S1 Fig. Viral DNA extraction method applied to the corneal tissue (adapted from the high pure viral nucleic acid kit workflow recommended by the manufacturer).**
(TIF)

**S2 Fig. Dendritic and geographic patterns obtained in the Active Storage Machine for each of the six corneas.** The infection pattern is shown at the top of each picture. Scale bar: 1000μm.
(TIF)

**S1 Table. Primary antibodies used for immunostaining on flat-mounted corneas and on cross-sections.**
(DOCX)

**S2 Table. Characteristics of the pairs of corneas, and main results obtained.**
(DOCX)

## Author Contributions

**Conceptualization:** Emilie Courrier, Marc Labetoulle, Philippe Gain, Gilles Thuret.

**Investigation:** Emilie Courrier, Corantin Maurin, Victor Lambert, Didier Renault, Sylvie Pillet, Fabien Forest, Chantal Perrache, Zhiguo He, Thibaud Garcin.

**Methodology:** Emilie Courrier, Didier Renault, Thomas Bourlet, Sylvie Pillet, Paul O. Verhoeven, Philippe Gain, Gilles Thuret.

**Supervision:** Antoine Rousseau, Philippe Gain, Gilles Thuret.

**Writing – original draft:** Emilie Courrier, Philippe Gain, Gilles Thuret.

**Writing – review & editing:** Antoine Rousseau, Marc Labetoulle.

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
