## [Decision Letter · Decision Letter 0]

28 Apr 2020

PONE-D-20-06801

Ex vivo model of Herpes Simplex virus Type I dendritic and geographic keratitis using a corneal active storage machine

PLOS ONE

Dear Pr. Thuret,

Thank you for submitting your manuscript to PLOS ONE. After careful consideration, we feel that it has merit but does not fully meet PLOS ONE’s publication criteria as it currently stands. Therefore, we invite you to submit a revised version of the manuscript that addresses the points raised during the review process.

The reviewers have requested minor changes, all of which can be addressed by making relatively minor changes in the text.  Please respond to all of the the reviewers' comments.  Importantly, carefully go over the manuscript for language issues.

We would appreciate receiving your revised manuscript by Jun 12 2020 11:59PM. To enhance the reproducibility of your results, we recommend that if applicable you deposit your laboratory protocols in protocols.io, where a protocol can be assigned its own identifier (DOI) such that it can be cited independently in the future. For instructions see: http://journals.plos.org/plosone/s/submission-guidelines#loc-laboratory-protocols

We look forward to receiving your revised manuscript.

Kind regards,

Nancy M Sawtell

Academic Editor

PLOS ONE

2. We note that you have a patent relating to material pertinent to this article. Please provide an amended statement of Competing Interests to declare this patent (with details including name and number), along with any other relevant declarations relating to employment, consultancy, patents, products in development or modified products etc. Please confirm that this does not alter your adherence to all PLOS ONE policies on sharing data and materials, as detailed online in our guide for authors http://journals.plos.org/plosone/s/competing-interests by including the following statement: "This does not alter our adherence to  PLOS ONE policies on sharing data and materials.” If there are restrictions on sharing of data and/or materials, please state these. Please note that we cannot proceed with consideration of your article until this information has been declared.

Reviewers' comments:

Reviewer's Responses to Questions

**Comments to the Author**

1. Is the manuscript technically sound, and do the data support the conclusions?

Reviewer #1: Yes

Reviewer #2: Yes

Reviewer #3: Yes

2. Has the statistical analysis been performed appropriately and rigorously? 

Reviewer #1: Yes

Reviewer #2: Yes

Reviewer #3: Yes

3. Have the authors made all data underlying the findings in their manuscript fully available?

Reviewer #1: Yes

Reviewer #2: Yes

Reviewer #3: Yes

4. Is the manuscript presented in an intelligible fashion and written in standard English?

Reviewer #1: Yes

Reviewer #2: No

Reviewer #3: Yes

5. Review Comments to the Author

Reviewer #1: INTRODUCTION

P3, L72: Correct to “The diagnosis of HK …”

METHODS

Study design, in which pairs of donor eye s were used, is a strength of the project.

RESULTS

- Chronological images in Fig 3 and electron micrographs (Fig 7) are particularly well presented.

- Fig 4: Why does the To-PRO 3 counterstained nucleus in infected cells appear larger than in uninfected control cornea in these en face images? (Authors refer to this in P18 line 432)

- Fig 5 illustrates significant thickening of epithelium in ASM-stored corneas compared to uninfected control cornea. Authors should comment on this.

- Fig 5: What is the explanation for significant gB+ staining of keratocytes in OC corneas?

- Fig 7C: Complete absence of the sub-basal plexus nerves is very striking and commented upon in Discussion.

DISCUSSION

The authors’ arguments in favour of the attributes of their ASM model are accepted. I agree that the model may have value in screening new anti-HSV treatments. However the biologically important limitations of an ex vivo model must be even more clearly emphasised than in L599-605, in particular the absence of any host immune response to virus infection. In HSV infection this interaction is recognised to be of major clinical importance and indeed it is stromal inflammation resulting from recurrent HSV infection which leads to visual loss. There is no substitute for an in vivo model to investigate these aspects of viral infection. As it stands, the model described by the authors has no usefulness in evaluating anti-inflammatory or immunomodulatory interventions in herpetic keratitis.

Reviewer #2: This manuscript presents data related to the use of an active storage machine (ASM) for the storage of human corneas after removal post mortem. The authors have published the ability of ASM storage to extend the transplantable life of corneas (refs 23, 24). Here, the authors utilize those corneas not adequately preserved for human transplant to investigate their use as a model for herpes simplex virus corneal infection by comparing ASM stored corneas to corneas stored by a standard organ culture method. Overall, this is an interesting novel study and data presented support the authors’ conclusions. The limitation of this as a model for HSK is the absence of key cellular components involved in this inflammatory disease process. This should be emphasized.

There are language issues throughout the manuscript and although minor, these need to be addressed. Some but not all are noted below. A careful review of language use is needed.

Specific comments

Line 33

Abstract ..”physiological state sufficiently close to physiology since they…” reword so the meaning of this statement is clearly stated.

The authors have not compared the physiological state of the corneas stored in ASM to corneas just after removal or at least this is not reported here. Just how similar the ASM stored cornea physiology is to “normal” is not known or at least not presented here.

Methods

Line 115 might facilitate reader understanding to concisely state the usual clinical characteristics here

Line 116; include viral strain and titer utilized

Line 130; please provide brief explanation as to how the scleral rim is used as a watertight seal

Line 299: why was cornea fixed in methanol and stored at 4C and not immediately in LN2 prior to DNA extraction

Not clear why investigators limited detection of HSV by the use of a single antibody. Would be strengthened by using additional anti HSV- antibody or antibody recognizing multiple viral proteins for enhancing sensitivity.

How many genomes are detected? 21 is equivalent to how many genomes? It is not clear whether these are relative numbers or represent actual genome copy numbers based on standards. This should be more clearly presented. Also, the sensitivity of the DNA recovery method should be demonstrated.

Lines 430-432. Were nuclei measured and correlated to another measure of CPE? if so, this data should be presented. If not, how was this conclusion made. As presented, this seems arbitrary. What is the data supporting enlarged nuclei = CPE. Are these syncytium?

Line 4565-56. What is the implication of this observation?

Fig 3 B This staining is not very convincing. Is it possible to show at higher power? What is the advantage of unfixed IHC?

Legend Fig 4. Please state the fixation status and type of fixation (if fixed) used.

language issues,

line 35 intended "for" not “to”

lines 39-42 rewording needed

line 46 damage not damages

line 49 “in” not "on"

line 57 “type” should be "types"

line 62 infancy is the first year of life. A broader term, perhaps early childhood?

Line 64 add reference

Line 81 “Even if” replace with "Although"

Line 102 “enzymatic equipment”. Not clear what this means

Line 171. Convert to a complete sentence, for example "Both were returned to 31C."

Line 174 awkward wording

Line 185 define PEEK

Line 190 what is meant by “after contamination…”

Line 194 “from” should be “for”

Line 221 After immunoabeling, corneas were removed….

Line 247 “excepted” should be “except”

Line 299 extracted “on” should be “from”

Lne 302 “realized” should be “carried out”. Nucleic acids “on” should be “from”

Line 305 “the number of viral DNA by cell”. This does not make sense

Line 308 “PCR” should be PCRs

Line 316 awkward wording

Line 319 should reword. “To our knowledge, dendritic lesions have not been previously reported in human cornea ex vivo. “

Line 516 ? ..”weakness epithelium” ?

Line 517-518 Please improve this sentence. “ ..closer to physiology than passive storage..? …….”real rehabilitation” what is meant? “Corneas having undergone alterations “?

Line 520 What data support “very close to the normal state” ?? have RNA seq data been generated? Or is this restricted to visual assessment.

Line 563 “ …eliminated the blinking eyelids” add “by”

Line 574-575 something seems to be missing. “Since, this model…”

preposition use should be examined closely

Reviewer #3: The active storage machine is a smart technological development in corneal storage and restoration. It is wonderful to read how succesfull the authors have been at applying this technology in order to develop a solid ex vivo model for epithelial herpetic keratitis - very relevant achievement for further research in the fight against herpes simplex and other infectious micro-organisms. Equally important are the findings that neither the immune system nor epithelial innervation are essential to explain the dendritiform shape of typical lesions.

The experiments are well set up with methods and results clearly and extensively presented. Interpretation of the results is sound and well founded.

6. PLOS authors have the option to publish the peer review history of their article (what does this mean?). If published, this will include your full peer review and any attached files.

Reviewer #1: Yes: Daniel F P LARKIN

Reviewer #2: No

Reviewer #3: No

---

## [Author Response · Author response to Decision Letter 0]

29 Jun 2020

June 10th 2020

Dear Ms. Sawtell,

 Please find attached a revised version of our article entitled “Ex vivo model of herpes simplex virus type I dendritic and geographic keratitis using a corneal active storage machine”.

 We have carefully considered the journal's requirements and responded to all the points addressed by the reviewers.

 As per your instructions, all amendments in the R1 version are indicated in our point-by-point response and are marked in red in the article.

 We greatly hope that this new version will meet the reviewers' expectations and comply with your editorial policy. 

Sincerely yours,

Prof Gilles Thuret (corresponding author)

Saint-Etienne University Hospital

"Corneal Graft Biology, Engineering, and Imaging" Laboratory EA 2521, SFR143

Faculty of Medicine

Saint-Etienne

France

PONE-D-20-06801

Ex vivo model of Herpes Simplex virus Type I dendritic and geographic keratitis using a corneal active storage machine

PLOS ONE

Dear Pr. Thuret,

Thank you for submitting your manuscript to PLOS ONE. After careful consideration, we feel that it has merit but does not fully meet PLOS ONE’s publication criteria as it currently stands. Therefore, we invite you to submit a revised version of the manuscript that addresses the points raised during the review process.

The reviewers have requested minor changes, all of which can be addressed by making relatively minor changes in the text. Please respond to all of the reviewers' comments. Importantly, carefully go over the manuscript for language issues.

We would appreciate receiving your revised manuscript by Jun 12 2020 11:59PM. To enhance the reproducibility of your results, we recommend that if applicable you deposit your laboratory protocols in protocols.io, where a protocol can be assigned its own identifier (DOI) such that it can be cited independently in the future. For instructions see: http://journals.plos.org/plosone/s/submission-guidelines#loc-laboratory-protocols

• A rebuttal letter that responds to each point raised by the academic editor and reviewer(s). This letter should be uploaded as separate file and labeled 'Response to Reviewers'.

• A marked-up copy of your manuscript that highlights changes made to the original version. This file should be uploaded as separate file and labeled 'Revised Manuscript with Track Changes'.

• An unmarked version of your revised paper without tracked changes. This file should be uploaded as separate file and labeled 'Manuscript'.

We look forward to receiving your revised manuscript.

Kind regards,

Nancy M Sawtell

Academic Editor

PLOS ONE

2. We note that you have a patent relating to material pertinent to this article. Please provide an amended statement of Competing Interests to declare this patent (with details including name and number), along with any other relevant declarations relating to employment, consultancy, patents, products in development or modified products etc. Please confirm that this does not alter your adherence to all PLOS ONE policies on sharing data and materials, as detailed online in our guide for authors http://journals.plos.org/plosone/s/competing-interests by including the following statement: "This does not alter our adherence to PLOS ONE policies on sharing data and materials.” If there are restrictions on sharing of data and/or materials, please state these. Please note that we cannot proceed with consideration of your article until this information has been declared.

Reviewers' comments: 

In blue: point-by-point answers to the reviewers’ comments.

Comments to the Author

1. Is the manuscript technically sound, and do the data support the conclusions?

Reviewer #1: Yes

Reviewer #2: Yes

Reviewer #3: Yes

2. Has the statistical analysis been performed appropriately and rigorously?

Reviewer #1: Yes

Reviewer #2: Yes

Reviewer #3: Yes

3. Have the authors made all data underlying the findings in their manuscript fully available?

Reviewer #1: Yes

Reviewer #2: Yes

Reviewer #3: Yes

4. Is the manuscript presented in an intelligible fashion and written in standard English?

Reviewer #1: Yes

Reviewer #2: No

Reviewer #3: Yes

We have considered the corrections advised by Reviewer 2, and the manuscript has now been reviewed by a native English speaker.

5. Review Comments to the Author

Reviewer #1: INTRODUCTION

P3, L72: Correct to “The diagnosis of HK …”. 

Done. 

METHODS

Study design, in which pairs of donor eyes were used, is a strength of the project.

Thank you for this comment.

RESULTS

- Chronological images in Fig 3 and electron micrographs (Fig 7) are particularly well presented. Thank you.

- Fig 4: Why does the To-PRO 3 counterstained nucleus in infected cells appear larger than in uninfected control cornea in these en face images? (Authors refer to this in P18 line 432). 

All cellular manifestations of a viral infection are called the cytopathogenic effect. In HSV-1, this effect is most notably manifested by an increase in the volume of the nucleus (intranuclear viral replication) and by the cells becoming rounded. The nucleus of cells infected by the virus therefore appears wider than that of uninfected cells. In Figure 4, this cytopathogenic effect is marked by white arrows, and the legend states: "The white arrows show the rounded infected cells exhibiting the HSV-1 induced cytopathogenic effect with a higher nuclear-cytoplasmic ratio." 

We have also added (lines 424-426): “… a higher nuclear-cytoplasmic ratio due to nuclear replication of HSV-1. The CPE observed in corneal epithelial cells is similar to that observed in HSV-1 infected fibroblast cell culture (PMID :6100717).” 

- Fig 5 illustrates significant thickening of epithelium in ASM-stored corneas compared to uninfected control cornea. Authors should comment on this.

The thickness of corneal epithelium in the ASM was greater than in passive OC for two reasons: 

1/ The ASM can regenerate an epithelium with more layers than in passive OC 

(Guindolet D, Crouzet E, He Z, Herbepin P, Perrache C, Garcin T, Gauthier AS, Forest F, Peoc'h M, Gain P, Gabison E, Thuret G. Epithelial regeneration in human corneas preserved in an active storage machine. Translational Vision Science & Technology 2020;in press.)

2/ In this multilayered epithelium, the cytopathogenic effect further increased the difference between the ASM and OC.

This explanation has now been added to the figure legend (lines 411-414). 

- Fig 5: What is the explanation for significant gB+ staining of keratocytes in OC corneas? 

As the epithelium of OC-stored corneas was more fragile and had fewer layers, we suppose that the virus caused faster desquamation of the entire epithelium, passed through Bowman's layer and infected the anterior keratocytes. Indeed, as mentioned in lines 452-454, in the corneas that had no epithelium at the end of the experiment, we observed viral particles directly enveloped on Bowman's layer. This means the virus moves through the cell layers as in the clinic. 

We have now added this explanation to the Discussion (lines 532-533).

- Fig 7C: Complete absence of the sub-basal plexus nerves is very striking and commented upon in Discussion.

Yes, very striking indeed.

DISCUSSION

The authors’ arguments in favour of the attributes of their ASM model are accepted. I agree that the model may have value in screening new anti-HSV treatments. However the biologically important limitations of an ex vivo model must be even more clearly emphasised than in L599-605, in particular the absence of any host immune response to virus infection. In HSV infection this interaction is recognised to be of major clinical importance and indeed it is stromal inflammation resulting from recurrent HSV infection which leads to visual loss. There is no substitute for an in vivo model to investigate these aspects of viral infection. As it stands, the model described by the authors has no usefulness in evaluating anti-inflammatory or immunomodulatory interventions in herpetic keratitis.

You are right: our model does not take account of the host immune response. We feel that we sufficiently discussed this point (lines 596-602). However, we have now added lines 602-604 to say that our model, in its current state, is of no use for evaluating anti-inflammatory or immunomodulatory interventions in herpetic keratitis. 

But the ASM could also be used to test cytokine cocktails or the reaction of immune cells injected into the stroma.

Reviewer #2: This manuscript presents data related to the use of an active storage machine (ASM) for the storage of human corneas after removal post mortem. The authors have published the ability of ASM storage to extend the transplantable life of corneas (refs 23, 24). Here, the authors utilize those corneas not adequately preserved for human transplant to investigate their use as a model for herpes simplex virus corneal infection by comparing ASM stored corneas to corneas stored by a standard organ culture method. Overall, this is an interesting novel study and data presented support the authors’ conclusions. The limitation of this as a model for HSK is the absence of key cellular components involved in this inflammatory disease process. This should be emphasized.

This limitation was also highlighted by Reviewer 1. We have now emphasized the biologically important limitations of the ex vivo model, in particular the absence of any host immune response to virus infection, in the Discussion, stating that the model described here is of no use for evaluating anti-inflammatory and immunomodulatory interventions in herpetic keratitis (lines 602-604).

There are language issues throughout the manuscript and although minor, these need to be addressed. Some but not all are noted below. A careful review of language use is needed.

The language issues noted below have been addressed, and the manuscript has now been reviewed by a native English speaker.

Specific comments

Line 33

Abstract ..”physiological state sufficiently close to physiology since they…” reword so the meaning of this statement is clearly stated.

The authors have not compared the physiological state of the corneas stored in ASM to corneas just after removal or at least this is not reported here. Just how similar the ASM stored cornea physiology is to “normal” is not known or at least not presented here.

This was indeed not the purpose of this article. A comparison of the epithelium of ASM-stored corneas and fresh corneas procured very soon after death or those procured in theater during certain penetrating keratoplasties has just been published (Guindolet et al, TVST 2020, in press).

We have amended the abstract per the reviewer's recommendations: 

“Nevertheless, the two current corneal storage methods (hypothermia and organ culture (OC)) do not preserve corneas in good physiological condition, as they…”

Methods

Line 115 might facilitate reader understanding to concisely state the usual clinical characteristics here. 

Thank you. We have now added on line 113: “(dendritic or geographic ulcers)”

Line 116; include viral strain and titer utilized. 

This information was already in the manuscript (lines 159-160 in the initial reviewed version), but we have now also added it earlier in the paper (lines 113-114).

Line 130; please provide brief explanation as to how the scleral rim is used as a watertight seal

The epithelial chamber and the endothelial chamber are hermetically separated by the cornea itself, the sclera being secured by a clamping ring. We have now made this addition (line 126): “Briefly, the cornea was tightly secured to the ASM base, using the scleral rim as a watertight seal, compressed by a clamping ring, to separate the epithelial and endothelial chambers”

Line 299: why was cornea fixed in methanol and stored at 4C and not immediately in LN2 prior to DNA extraction

Methanol does not interfere with the Roche extraction kit, especially as we amplified a cell gene, which shows the integrity of nucleic acids. The storage mode does not therefore interfere with the PCR. Methanol was chosen because it is also a fixative, used here before immunostaining with E-cadherin.

Not clear why investigators limited detection of HSV by the use of a single antibody. Would be strengthened by using additional anti HSV- antibody or antibody recognizing multiple viral proteins for enhancing sensitivity.

We did not consider it necessary to add an additional antibody, given that the present one showed good specificity after in vitro testing on HSV-1 infected and uninfected cell culture, but also on HSV-1 infected and uninfected corneas (data not shown). Moreover, we had other ways of detecting the presence of infection (morphological alterations of cells, PCR, MET).

How many genomes are detected? 21 is equivalent to how many genomes? It is not clear whether these are relative numbers or represent actual genome copy numbers based on standards. This should be more clearly presented. Also, the sensitivity of the DNA recovery method should be demonstrated. 

We used a kit enabling estimation of the actual viral load of the HSV-1 virus based on quantification standards. In addition, we estimated the number of cells per sample using a housekeeping gene (HPRT1). We were thus able to deduce the viral load/cell, i.e. the actual number of copies of the viral genome/cell. The viral load was normalized relative to the number of cells so as to eliminate the variability of the cornea sample sizes. We did the PCR twice on each sample, and kept the average viral load of each one. We have now added these details in Materials and Methods (lines 299 and 305). 

The figures given in Results (lines 493-494) thus correspond to the "actual" viral load per cell, and are not relative values. The "genome copies/1000cells" unit has been added (lines 493-494). 

We have also added a reference for the relative quantification of the HSV-1 genome relative to the HPRT1 cell gene (Frobert E, Billaud G, Casalegno JS, Eibach D, Goncalves D, Robert JM, Lina B, Morfin F. The clinical interest of HSV1 semi-quantification in bronchoalveolar lavage. J Clin Virol 2013;58:265-268) on line 305. 

Lines 430-432. Were nuclei measured and correlated to another measure of CPE? if so, this data should be presented. If not, how was this conclusion made. As presented, this seems arbitrary. What is the data supporting enlarged nuclei = CPE. Are these syncytium?

Thank you for your comment. We did not measure the nuclei or correlate them with another measure of the CPE. It was clear that the infected cells (gB+ staining) had a larger nucleus than the uninfected ones. Reviewer 1 also asked us to discuss this point. In the case of HSV-1, the CPE notably results in increased nucleus volume due to nuclear replication of the virus; and in cell rounding, which we observed here relative to uninfected cells (remark added in lines 424-426). This CPE looks like the one commonly observed in HSV-1 infected fibroblast cultures (Hsiung GD, Landry ML, Mayo DR, Fong CK. Laboratory diagnosis of herpes simplex virus type 1 and type 2 infections. Clin Dermatol 1984;2:67-82.). We have now added this reference (line 426).

Line 455-56. What is the implication of this observation?

This observation seems to confirm that in the case of organ-cultured corneas (which have a more fragile epithelium with fewer cell layers), the virus reaches Bowman's layer more quickly, which would explain the larger number of infected keratocytes. Indeed, as mentioned on lines 452-454, in the case of the corneas without epithelium at the end of the experiment, we observed viral particles enveloped directly on Bowman's layer. We were also able to observe, using TEM, some infected keratocytes where the enveloped viral particles were found in the cytoplasm.

We have now discussed this comment in the Discussion (lines 532-533).

Furthermore, enveloped viral particles were observed in the keratocytes, as mentioned (lines 454-456). Finding the virus deeper in the stroma is interesting, as it reproduces the infectious phase of stromal keratitis. Of course, the present model cannot reproduce immune stromal keratitis, but adding immune effectors to the ASM may be envisaged to try to reproduce this dreadful condition.

This finding is now discussed (lines 533-537).

Fig 3 B This staining is not very convincing. Is it possible to show at higher power? What is the advantage of unfixed IHC?

The luminosity and contrast of the fluorescent images in the figure have now been improved.

Legend Fig 4. Please state the fixation status and type of fixation (if fixed) used.

Done.

• language issues,

line 35 intended "for" not “to” OK.

lines 39-42 rewording needed Done.

line 46 damage not damages OK.

line 49 “in” not "on" OK.

line 57 “type” should be "types" OK.

line 62 infancy is the first year of life. A broader term, perhaps early childhood? OK.

Line 64 add reference We have added a reference (PMID: 11861220). 

Line 81 “Even if” replace with "Although" OK.

Line 102 “enzymatic equipment”. Not clear what this means

This has now been changed to “expression of NA+/K+ ATPase”.

Line 171. Convert to a complete sentence, for example "Both were returned to 31C." OK.

Line 174 awkward wording Done (lines 170-171). 

Line 185 define PEEK OK.

Line 190 what is meant by “after contamination…”. Contamination refers to HSV-1 infection. The term has been replaced.

Line 194 “from” should be “for” OK.

Line 221 After immunoabeling, corneas were removed….OK.

Line 247 “excepted” should be “except” OK.

Line 299 extracted “on” should be “from” OK.

Lne 302 “realized” should be “carried out”. Nucleic acids “on” should be “from” OK.

Line 305 “the number of viral DNA by cell”. This does not make sense

We were referring to the estimated viral load as a number of copies of the viral genome/1000cells. This has now been clarified (line 299).

Line 308 “PCR” should be PCRs OK.

Line 316 awkward wording Done (line 312).

Line 319 should reword. “To our knowledge, dendritic lesions have not been previously reported in human cornea ex vivo. “ OK.

Line 516 ? ..”weakness epithelium” 

We have now replaced this by “… causes further deterioration of the already impaired epithelium” (lines 506-507).

Line 517-518 Please improve this sentence. “ ..closer to physiology than passive storage..? …….”real rehabilitation” what is meant? “Corneas having undergone alterations “?

We wish to point out that the ASM improves the condition of corneas that are normally unusable, so we feel that the term "rehabilitate" is appropriate.

We have now amended this sentence as follows:

"By restoring corneas to a far more physiological environment than passive OC, our ASM rehabilitates corneas discarded by eye banks, which have a severely degraded epithelium."

Line 520 What data support “very close to the normal state” ?? have RNA seq data been generated? Or is this restricted to visual assessment.

Not yet! But even more importantly than the transcriptome, which is not always translated into proteins, we compared the histology and several characteristic proteins of the humal corneal epithelium and of the limbus. This is described in a paper in press in TVST. We have added this reference, which should be available very soon. 

Line 563 “ …eliminated the blinking eyelids” add “by” This has been reworded (lines 560-561). 

Line 574-575 something seems to be missing. “Since, this model…” The sentence has been reworded (line 572).

preposition use should be examined closely

Reviewer #3: The active storage machine is a smart technological development in corneal storage and restoration. It is wonderful to read how succesfull the authors have been at applying this technology in order to develop a solid ex vivo model for epithelial herpetic keratitis - very relevant achievement for further research in the fight against herpes simplex and other infectious micro-organisms. Equally important are the findings that neither the immune system nor epithelial innervation are essential to explain the dendritiform shape of typical lesions.

The experiments are well set up with methods and results clearly and extensively presented. Interpretation of the results is sound and well founded.

Thank you for your comments!

---

## [Editor Report · Decision Letter 1]

1 Jul 2020

Ex vivo model of Herpes Simplex virus Type I dendritic and geographic keratitis using a corneal active storage machine

PONE-D-20-06801R1

Dear Dr. Thuret,

We’re pleased to inform you that your manuscript has been judged scientifically suitable for publication and will be formally accepted for publication once it meets all outstanding technical requirements.

Kind regards,

Nancy M Sawtell

Academic Editor

PLOS ONE
---

## [Editor Report · Acceptance letter]

9 Jul 2020

PONE-D-20-06801R1 

Ex vivo model of Herpes Simplex virus Type I dendritic and geographic keratitis using a corneal active storage machine 

Dear Dr. Thuret:

I'm pleased to inform you that your manuscript has been deemed suitable for publication in PLOS ONE. Congratulations! Your manuscript is now with our production department. 

Kind regards, 

on behalf of

Dr. Nancy M Sawtell 

Academic Editor

PLOS ONE